# A Comparative Analysis of Two Automated Quantification Methods for Regional Cerebral Amyloid Retention: PET-Only and PET-and-MRI-Based Methods

**DOI:** 10.3390/ijms25147649

**Published:** 2024-07-12

**Authors:** Sunghwan Kim, Sheng-Min Wang, Dong Woo Kang, Yoo Hyun Um, Eun Ji Han, Sonya Youngju Park, Seunggyun Ha, Yeong Sim Choe, Hye Weon Kim, Regina EY Kim, Donghyeon Kim, Chang Uk Lee, Hyun Kook Lim

**Affiliations:** 1Department of Psychiatry, College of Medicine, Yeouido St. Mary’s Hospital, The Catholic University of Korea, 222 Banpo-daero, Seocho-gu, Seoul 06591, Republic of Korea; 2Department of Psychiatry, College of Medicine, Seoul St. Mary’s Hospital, The Catholic University of Korea, 222 Banpo-daero, Seocho-gu, Seoul 06591, Republic of Korea; 3Department of Psychiatry, St. Vincent’s Hospital, College of Medicine, The Catholic University of Korea, 222 Banpo-daero, Seocho-gu, Seoul 06591, Republic of Korea; 4Division of Nuclear Medicine, Department of Radiology, Yeouido St. Mary’s Hospital, College of Medicine, The Catholic University of Korea, 222 Banpo-daero, Seocho-gu, Seoul 06591, Republic of Korea; 5Division of Nuclear Medicine, Department of Radiology, Seoul St. Mary’s Hospital, The Catholic University of Korea, 222 Banpo-daero, Seocho-gu, Seoul 06591, Republic of Korea; 6Research Institute, Neurophet Inc., Seoul 06234, Republic of Koreareginaeunyoungkim@neurophet.com (R.E.K.);; 7CMC Institute for Basic Medical Science, The Catholic Medical Center of The Catholic University of Korea, 222 Banpo-daero, Seocho-gu, Seoul 06591, Republic of Korea

**Keywords:** amyloid retention, automated quantification, PET-and-MRI-based method, Alzheimer’s disease

## Abstract

Accurate quantification of amyloid positron emission tomography (PET) is essential for early detection of and intervention in Alzheimer’s disease (AD) but there is still a lack of studies comparing the performance of various automated methods. This study compared the PET-only method and PET-and-MRI-based method with a pre-trained deep learning segmentation model. A large sample of 1180 participants in the Catholic Aging Brain Imaging (CABI) database was analyzed to calculate the regional standardized uptake value ratio (SUVR) using both methods. The logistic regression models were employed to assess the discriminability of amyloid-positive and negative groups through 10-fold cross-validation and area under the receiver operating characteristics (AUROC) metrics. The two methods showed a high correlation in calculating SUVRs but the PET-MRI method, incorporating MRI data for anatomical accuracy, demonstrated superior performance in predicting amyloid-positivity. The parietal, frontal, and cingulate importantly contributed to the prediction. The PET-MRI method with a pre-trained deep learning model approach provides an efficient and precise method for earlier diagnosis and intervention in the AD continuum.

## 1. Introduction

Alzheimer’s disease (AD) is the most common cause of dementia and thus one of the most burdensome diseases globally [1]. Individuals affected by the disease exhibit a wide range of symptoms, such as memory impairment, behavioral disturbance, and psychological symptoms [2,3] or may even lack any clinical symptoms, which is called preclinical AD [4]. Recently, the importance of biomarkers measuring the deposition of β-amyloid (Aβ) plaques in the brain—a central pathological hallmark of AD—has increasingly been recognized [5,6]. According to the 2018 National Institute on Aging–Alzheimer’s Association (NIA-AA) framework, AD is defined based on postmortem examination or biomarkers, rather than clinical symptoms [7]. Notably, the relevance of early identification of Aβ deposition in individuals without cognitive impairment is underscored by evidence showing that the Aβ pathology may begin up to twenty years before noticeable cognitive deterioration emerges [8,9,10,11]. Moreover, recently developed disease-modifying treatments (DMTs) such as lecanemab and donanemab, which target and alleviate cerebral Aβ deposition, have highlighted the crucial need for incorporating biomarkers into the diagnostic and therapeutic frameworks used in real-world clinics for the geriatric population [12,13].

Two nonequivalent biomarkers, such as fluid (cerebrospinal fluid (CSF) or blood) and imaging markers, are recommended to assess the pathologic Aβ burden, even in the early course of AD disease trajectories [7]. Among these, amyloid positron emission tomography (PET) imaging markers possess distinct advantages over fluid biomarkers. Amyloid PET scans using radioactive tracers (details are described in the Appendix A) that bind to fibrillar amyloid aggregate in the brain and allow physicians to conduct in vivo visualization of the spatial distribution and regional quantification of Aβ protein deposits in the brain [14,15]. It reflects a more cumulative effect of Aβ and the static status of AD compared to fluid biomarkers [16,17]. In contrast, CSF or blood-based biomarkers could be affected by medical comorbidities or ethnicities [18,19], which is not observed with neuroimaging markers. Blood-based biomarkers, in particular, are subject to the influences of physical conditions, which may lead to diminished test–retest reliability [20,21]. Therefore, amyloid PET is increasingly considered a preferred modality for the precise clinical diagnosis of AD.

When using amyloid PET, a visual reading of scanned images by nuclear medicine physicians is deemed the most accurate method to determine the positivity of Aβ pathology [22]. However, this method requires considerable time from experienced readers and introduces potential variability within or between raters [23]. An operator-independent methodology to reduce this bias is quantitative assessments of the regional Aβ burden [24]. The standardized uptake value ratio (SUVR), a ratio of Aβ uptake values between predefined atlas regions and a reference region (e.g., cerebellum or pons), is one of the most widely used quantification methods for measuring regional Aβ retention in the brain [25]. A semi-automated process for binary classification is possible by thresholding the mean of SUVRs [26]. Recently, various cut-off values for binary classification of Aβ retention have been suggested but they can vary with the selection of reference region, tracers, or preprocessing methods [27,28,29,30]. Despite these obstacles, regional SUVRs have presented excellent features in predicting prognosis [17,31]; thus, fast and accurate numeric assessment for regional Aβ could help diagnose the presence of AD pathology [32,33].

Several commercially available pieces of software such as Syngo.via VB60A (Siemens, Inc. Erlangen, Germany) or BRASS (https://www.hermesmedical.com/our-software/neurology/ accessed on 9 July 2024, Hermes Medical Solution) have gained regulatory approval from both the FDA (Food and Drug Administration) and EMA (European Medicines Agency) [34]. These platforms facilitate fast and automatic quantification of regional Aβ retention and assist physician’s decision making. Typically, such software spatially normalizes the PET images into standard space using the Montreal Neurological Institute (MNI) templates and extracts uptake values of each region-of-interest (ROI) based on the pre-defined atlas (e.g., Automated Anatomical Labeling (AAL) atlas). Nevertheless, these atlas-based methods that rely solely on PET imaging may yield suboptimal SUVR calculations in the case of severe brain atrophy [35]. Alternative methods, including PETSurfer [36,37,38] and PMOD [39], could mitigate this issue by using structural magnetic resonance images (MRI) to co-register with PET images and augment anatomical information; however, they require a substantially longer time to calculate the end results.

To address this, a deep learning-based tool for PET and MRI segmentation in individual spaces using a pre-trained model has been recently introduced [40]. This approach provides more accurate and rapid production of SUVRs for each region based on subject-specific ROIs than previous methods; thus, it could detour the hurdles including the risk of improper registration or extensive time consumption for registration. In particular, equivocal results are frequently reported by visual readings in preclinical samples [41] and a deep learning-based method could have clinical utility for such cases [42]. While these advancements are promising, few studies have performed head-to-head comparisons of the performance between different quantification methods.

The purpose of this study is to evaluate the discriminative ability to identify Aβ-positivity using two established pieces of quantification software—the conventional PET-only method and deep learning-based method using PET and MRI—in a large population including cognitively normal or impaired samples. We hypothesized that the deep learning-based segmentation model would represent superior classification performance for Aβ-positivity because of its ability to augment high-resolution topographic information.

## 2. Results

### 2.1. Demographics and Clinical Characteristics

The demographic and clinical characteristics of study participants are represented in Table 1. The Aβ positive and negative groups consisted of 529 and 651 individuals, respectively. There was a statistically significant difference in age between the two groups (*t* = −6.49, *p* < 0.001) but no significant differences in other variables such as educational years or the proportion of females.

### 2.2. Consistency of Amyloid PET SUVR between Two Quantification Methods

The SUVRs derived from two different methods showed statistically significant positive correlations across all five ROIs in all participants, as well as within subgroups categorized by their clinical diagnoses (all *p* < 0.001; see Figure 1). For all participants, global SUVRs between the two methods showed a significant correlation (r = 0.931). In addition, regional SUVRs showed the strongest correlation in the temporal region (r = 0.973), followed by the cingulate (r = 0.965), frontal (r = 0.951), parietal (r = 0.948), and striatal (r = 0.840) areas between two methods. Subgroup analysis revealed a similar correlation coefficient for global SUVRs among CU, MCI, and DE groups, with coefficients of 0.887, 0.934, and 0.936, respectively. Among five regions, the cingulate, frontal, and temporal areas exhibited relatively stronger correlations between two quantification methods with coefficients ranging from 0.942 to 0.972, while the parietal and striatal regions represented weaker associations whose coefficients ranged from 0.830 to 0.944. The Bland–Altman plot is also represented in the Appendix A.

### 2.3. Difference in Amyloid PET SUVR between Two Quantification Methods

The distribution of SUVR measures and the differences between the two methods for each group are displayed in Figure 2. The difference in global SUVRs between the two methods was statistically significant for the CU (*t* = 5.270, *p* < 0.001) and MCI (*t* = 2.371, *p* = 0.018) groups but not for the DE (*t* = 0.128, *p* = 0.89) group (Figure 2A and Appendix A). Significant differences in the mean frontal and parietal SUVRs between the two methods were observed across all three groups (Figure 2B,D).

The average RMSE between global SUVRs from the two methods were 0.061 ± 0.033 for CU, 0.053 ± 0.037 for MCI, and 0.054 ± 0.040 for DE groups, respectively (Figure 2G and Appendix A). Among the regional SUVRs of five ROIs, the average RMSE from the two methods ranged from 0.033 to 0.097 across all three subgroups. The RMSE from the two methods for the frontal lobe showed significant group differences between each pair of the CU, MCI, and DE groups; the RMSE for the parietal lobe represented a significant difference between CU and DE and between MCI and DE (Figure 2H,J).

### 2.4. Predicting Visual Reads for Aβ-Positivity Using Regional SUVRs

The comparison results of LR models from performance measurement using two quantification methods are presented in Table 2 and Figure 3. The AUROC of SCALE PET for all participants was 0.956, which is higher than that of Syngo.via, 0.947. The accuracy, sensitivity, and F1 score were higher in SCALE PET. Considering subgroups, all metrics showed better capability of classifying Aβ-positivity with SCALE PET in the CU and MCI group and AUROC and specificity were better with Syngo.via in the DE group. The difference in AUROC was statistically significant in all participants (*p* = 0.002, DeLong’s test) and the CU subgroup (*p* = 0.013, DeLong’s test) but not significant in the MCI subgroup (*p* = 0.068, DeLong’s test) or the DE subgroup (*p* = 0.323, DeLong’s test).

The coefficients for the two LR models trained with the whole dataset using different quantification methods exhibited similar patterns (Figure 4). In the Syngo.via model, the parietal lobe (4.74) had the highest importance, followed by the cingulate (4.69), temporal lobe (3.90), frontal lobe (3.90), and striatal area (1.38). In the SCALE PET model, the parietal area also had the greatest importance (4.76), followed by the cingulate (4.62), frontal (4.18), temporal (2.21), and striatal (1.54) areas.

We also conducted the same analysis using a global SUVR as the input or an SVM as an alternative model (Appendix A). The results were similar to those of LR models using regional SUVRs. Additionally, the optimal cutoff values for global SUVR composites were calculated and summarized in Appendix A.

## 3. Discussion

To our knowledge, this is the first study comparing the conventional PET-only method with a deep learning-based method using both PET and MRI for automatic quantification of Aβ deposition in a large and balanced sample. The SCALE PET, which integrates PET and MRI into a pre-trained deep learning model for segmentation, and Syngo.via, which uses only PET images for registration, showed consistent prediction for SUVRs with high correlation coefficients and low levels of difference. However, there were significant differences in the ability to predict Aβ-positivity in CU groups. The SUVRs of parietal, cingulate, and frontal areas were importantly used in each logistic regression model.

The predicted SUVR values of PET images from our dataset, obtained using both the PET-only method in standard space and the PET-MRI method in an individual’s native space, generally demonstrated high correlation and minimal error but discrepancies were observed in the parietal, frontal, and striatal regions. When using the PET-only method and registering patients’ PET images to MNI space, some errors may occur in cortical areas such as the frontal and parietal regions or smaller regions like the striatum [43,44]. Especially, these errors are likely to be increased in cases of brain atrophy, such as with MCI or dementia [35]. However, utilizing both PET and MRI and segmenting in native space rather than standard space could reduce these potential errors, enabling more accurate extraction of SUVR values [45].

In the LR model, to predict visual reading by using regional SUVRs as inputs, values extracted via the PET-MRI method significantly improved the performance of the model, with the parietal, frontal, and cingulate lobes being particularly important for predicting Aβ-positivity. This finding was consistent across all subgroups. Previous amyloid imaging and pathology studies have indicated that these regions are among the earliest sites of Aβ deposition [46,47]. Aβ first accumulates in the precuneus, cingulate, and orbitofrontal areas before spreading to the somatosensory, lingual, and pericalcarine gyri in later stages. Recent research has identified subgroups showing initial Aβ retention starting in the frontal or parietal regions [48,49]. Another study showed that frontal and lateral parietal deposition were important predictors of future conversion to Aβ-positive among Aβ-negative individuals [50]. Notably, the frontal and parietal areas are functionally interconnected, forming the frontoparietal network. This frontoparietal and limbic network containing the cingulate cortex is positively associated with the Aβ burden in cognitively normal older adults, as observed in a previous study [51]. Furthermore, inter-network connectivity between default mode and frontoparietal networks negatively mediates the association between global Aβ level and episodic memory function [52]. Considering that the frontal, parietal, and cingulate areas are heavily linked to the medial temporal lobes (MTL) [53,54], which are sites for early tau pathology aggregation [55], Aβ deposition in these regions could play an important role in the early course of the AD continuum in non-demented populations.

The superior performance of the PET-MRI method in cognitively normal subjects has significant implications. This is because the importance of early diagnosis is increasing in the AD field, where diagnostic frameworks based on biomarkers are evolving and DMTs are being introduced. DMTs were approved for use not at the dementia stage, where neuronal damage has already advanced, but at the MCI or mild dementia stage, and only in patients who have been confirmed to be Aβ-positive [13,56]. Recently, researchers have argued that the indications for DMTs should be extended to preclinical AD [57] based on the A-T-N (Amyloid-tau-neurodegeneration) hypothesis [10]. Considering the pathological characteristics where accumulation of tau protein and neurodegeneration progress after Aβ accumulation in the brain, it is believed that more aggressive treatment at the stage where Aβ begins to accumulate can prevent cognitive decline by halting the progression to the next pathological stage [58,59]. To administer treatment from the early stages of AD, even before symptoms appear, it is essential to identify amyloid-beta positivity in the normal population to select the appropriate subjects for treatment [60]. Therefore, the PET-MRI method, which can better diagnose AD in cognitively unimpaired subjects, could play a crucial role in such proactive early treatment and might potentially replace the PET-only method in the future.

This study has several strengths. Firstly, it conducted a comparative analysis using a large matched sample of 1,180 participants. Secondly, we utilized regional SUVRs and logistic regression models to classify Aβ-positivity, instead of simple thresholding for global SUVR. Employing a multivariable model, rather than just binarizing a single composite score, allows for the incorporation of the spatial pattern of Aβ burden that may not be captured by global SUVR alone. Through 10-fold cross-validation and AUROC comparison, we could evaluate the discriminability of two sets of regional SUVR values from different quantification methods to identify Aβ positive and negative groups [61]. We also tested an LR model with the global SUVR and an SVM model with regional SUVR as alternative models, which showed similar accuracy to the regional SUVR LR model. Thirdly, a deep learning-based approach could address issues of time, cost, and accuracy. The PET-MRI method provides superior anatomical information, though time consumption for preprocessing such as co-registration is a concern. We used a pre-trained deep learning segmentation model to reduce time cost (<15 min) and increase the precision of SUVR calculation [40].

The limitations of our study include the following. Firstly, it is a single-center study, using only one reference ROI and tracer. Therefore, the generalizability of our results remains questionable. Multi-center studies using multiple tracers with different reference ROIs should be followed in the future. Secondly, the ROIs between the two methods did not precisely match, leading to potential bias from ROI selection. Sensitivity analysis with varying target ROIs should be conducted to address these issues.

## 4. Materials and Methods

### 4.1. Study Participants

A total of 1180 adults (median age 77, range 36–96 years) were included in this study. Participants voluntarily enrolled in the Catholic Aging Brain Imaging (CABI) database, which houses PET scans and clinical data of the individuals attending the outpatient clinic at the Catholic Brain Health Center, Yeouido St. Mary’s Hospital, The Catholic University of Korea (Seoul, Republic of Korea), from January 2018 to December 2023. The inclusion criteria were subjects older than 36 years with an amyloid PET scan and the Korean Version of the Consortium to Establish a Registry for Alzheimer’s Disease Assessment Packet (CERAD-K) result rated by trained psychologists. The CERAD-K battery assesses several cognitive domains including memory (word list memory/recall/recognition and constructional praxis recall), visuospatial construction (constructional praxis), language (15-item Boston Naming test), attention, and executive functions (verbal fluency) [62]. The exclusion criteria were individuals (i) having any psychiatric or neurological disorder except for dementia or (ii) unstable medical conditions.

The patients with mild cognitive impairment (MCI) groups met the following criteria: (1) informant reports for memory complaints; (2) at least −1.0 standard deviation (SD) below norms adjusted by age and education in one or more cognitive domains on the CERAD-K, (3) preserved activities of daily living; (4) global clinical dementia rating score (CDR) of 0.5; and (5) mild neurocognitive disorder according to the Diagnostic and Statistical Manual of Mental Disorders, fifth edition (DSM-5) [63]. Patients in the DE group (1) had global CDR scores greater than or equal to 1 and (2) major neurocognitive disorders proposed by the DSM-5. Subjects with CERAD-K scores exceeding −1.0 SD for all domains were classified as the cognitively unimpaired (CU) group. According to these criteria, 422 subjects were categorized as CU, 605 as MCI, and 153 as DE. Diagnoses for each participant were made by three geriatric psychiatrists (H.K.L., S.M.W., and S.K.). The use of the CABI dataset was approved by the institutional review board of Yeouido St. Mary’s Hospital, the Catholic University of Korea (no. SC23RISI0100).

The design of this study is a cross-sectional study based on the data extracted by retrospective chart review and the IRB granted waivers of consent for the participants. The demographic variables including age, sex, educational years, *APOE* ε4 carrier status, and cognitive test results from the CERAD battery were obtained from the CABI dataset and analyzed in our study.

### 4.2. Amyloid PET and MRI Image Acquisition

Static PET scans were acquired 90 min after intravenous injection of 185MBq of ^18^F-flutemetamol (FMM) for 20 min with Biograph 40 TruePoint (Siemens Medical Solutions, Erlangen, Germany). The matrix size was 256 × 256 × 175 and the voxel size was 1.3364 × 1.3364 × 3 mm^3^. Reconstruction of a static image was performed with the 2D-ordered subsets expectation–maximization algorithm with two iterations for 21 subsets. The CT scans were acquired for attenuation correction before the PET scans.

Structural T1-weighted images were scanned through the Siemens Skyra 3T scanner (Siemens Healthcare, Erlangen, Germany) and a 20-channel head and neck coil. The magnetization-prepared rapid gradient echo scan (MPRAGE) sequence was utilized with scanning parameters as follows: repetition time (TR) = 1860 ms, echo time (TE) = 25.3 ms, flip angle = 9°, field-of-view (FOV) = 224 × 224 mm, matrix size of 256 × 256, and 208 axial slices with a slice thickness of 1.0 mm. All acquired Digital Imaging and Communications in Medicine (DICOM) image files were anonymized and converted to the NIfTI format with “dcm2niix” software https://github.com/rordenlab/dcm2niix (accessed on 9 July 2024), [64]. The imaging acquisition process is illustrated in Appendix A.

### 4.3. Visual Analysis of Amyloid PET

Visual inspection and interpretation of PET images were conducted by two nuclear medicine physicians (E.J.H. and S.Y.P.). They evaluated the presence of cerebral Aβ retention by examining the uptake intensity in the gray matter compared to the white matter in six specific brain areas: the frontal lobes, lateral temporal lobes, inferolateral parietal lobes, posterior cingulate, precuneus, and striatum [15,22]. A scan was considered Aβ-positive if any of these regions exhibited equal or greater uptake in the gray matter relative to the adjacent white matter. The visual reading results were obtained independently of the quantification results and were used as a gold standard for Aβ-positivity [22].

### 4.4. Quantitative Assessment of Aβ Deposition

To quantify the regional and global level of Aβ retention in the brain, Syngo.via (Ver. VB20A, Siemens Medical Solutions Inc., Malvern, PA, USA) and SCALE PET (Ver.1.0.0, Neurophet Inc., Seoul, Republic of Korea) were utilized in this study. Siemens Syngo.via software VB60A [65], which is one of the widely used pieces of software, is based on Fleisher’s method to assess the SUVRs of 18F-Florbetapir for patients with AD dementia, MCI, and older healthy controls [26]. This software offers regional SUVRs for several ROIs in the AAL atlas: the cortex of the frontal lobe, temporal lobe, parietal lobe, precuneus, anterior, and posterior cingulate cortex (ACC and PCC, respectively), and striatum. Global SUVR is also provided by averaging the six cortical ROIs, excluding the striatum. SCALE PET, a recently developed PET image analysis engine, features precise automated quantification and rapid processing speeds by a deep learning-based approach for direct application in clinical practice [40]. Frontal, temporal, parietal, cingulate, and striatal SUVRs, as well as whole-brain global SUVR values, are calculated based on the Desikan–Killiany atlas and integrated according to the Alzheimer’s Disease Neuroimaging Initiative (ADNI)’s preprocessing method [66,67]. To directly compare Syngo.via with SCALE PET, the SUVRs of the ACC and PCC from Syngo.via were averaged and referred to as “cingulate,” while the SUVRs of the parietal and precuneus regions were averaged and referred to as “parietal” in this study. The pons was used as the reference ROI [68,69].

### 4.5. Statistical Analysis

Demographic and clinical characteristics were compared, employing independent *t*-tests and chi-square tests. To validate the consistency of amyloid PET SUVR between two quantification methods, Pearson’s correlation coefficient and root-mean-squared-error (RMSE) between SUVR values derived from two different methods were calculated for whole brain and five regional ROIs such as the frontal, temporal, parietal, and cingulate cortex, and striatum areas. To be specific, the sum of the squared differences between the SUVRs obtained from the two methods was divided by the number of samples in each group and the square root of the result was taken to calculate the RMSE.

To compare the predictability of the two methods, logistic regression (LR) models were fitted using two sets of regional SUVRs as input features and Aβ-positivity as a predicting variable, regarding visual reads as a gold standard (Figure 5). For each LR model, SUVRs derived from five ROIs—frontal, temporal, parietal, cingulate, and striatum—were utilized. We also fitted the LR model using global SUVR only and support vector machine (SVM) using regional SUVRs as alternative models (more detailed information about LR and SVM is described in the Appendix A).

Performance metrics of the models, such as accuracy, sensitivity, specificity, precision, and the area under the receiver operating characteristic curve (AUROC), were analyzed by 10-fold cross-validation using the function “cross_val_predict” implemented in the ‘scikit-learn’ package. Specifically, the whole dataset was split into 10 non-overlapping subsets and all individual samples within each subset were predicted for Aβ positivity using 10 different LR or SVM models trained with data not belonging to the corresponding subset to avoid overfitting. A model refitted with the whole dataset was used to calculate feature importance. Comparative analysis of AUROC employed DeLong’s test [70] and its recent implementation [71] with a statistical significance threshold of *p* < 0.05. All statistical analyses were performed with the ‘scipy’ package (version 1.10.0) and the construction of models and the calculation of feature importance were executed using the ‘scikit-learn’ package (version 1.1.3) in Python 3.10.9. The Python implementation of DeLong’s algorithm was available at the authors’ repository (https://github.com/yandexdataschool/roc_comparison accessed on 1 April 2024).

## 5. Conclusions

In this study, we compared the two established pieces of quantification software and demonstrated that PET-MRI methods with a pre-trained deep learning model could have a superior ability to identify Aβ-positivity. The parietal, frontal, and cingulate areas were important to predict Aβ-positivity, especially in non-demented adults. To detect the early AD continuum and intervene promptly, using the PET-MRI method would be beneficial to efficiently and accurately quantify Aβ accumulation in these regions.

## Figures and Tables

**Figure 1 ijms-25-07649-f001:**
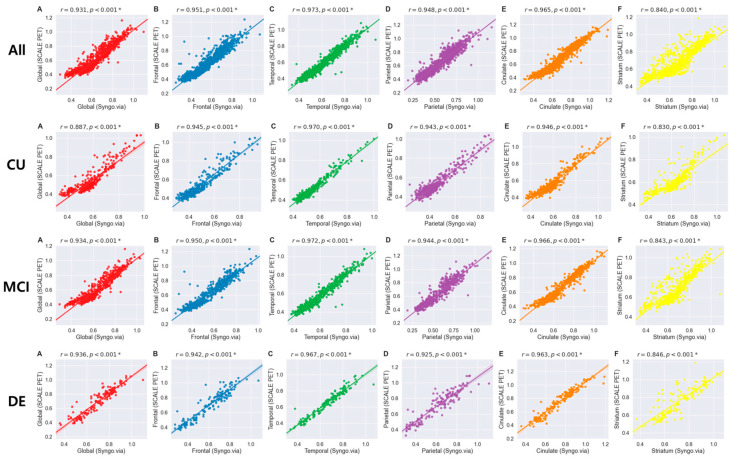
Correlation analysis results of regional amyloid burden between the deep learning-based PET-MRI (SCALE PET) and the PET-only (Syngo.via) methods for all (A–F in the **first row**, labeled “All”), CU (A–F in the **second row**, labeled “CU”), MCI (A–F in the **third row**, labeled “MCI”), and DE (A–F in the **fourth row**, labeled “DE) groups. Abbreviations: CU, cognitively unimpaired; MCI, mild cognitive impairment; DE, dementia. * denotes *p* < 0.05.

**Figure 2 ijms-25-07649-f002:**
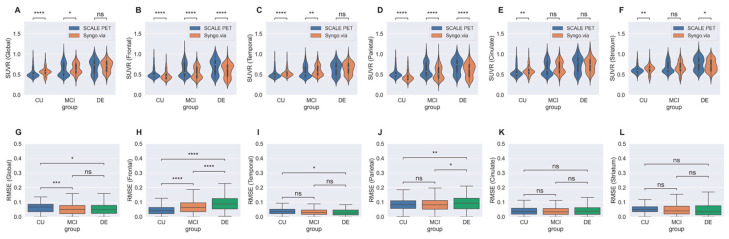
Distribution of SUVRs ((**A**–**F**) in the upper row) and root-mean-squared error ((**G**–**L**) in the lower row) between the deep learning-based PET-MRI (SCALE PET) and the PET-only (Syngo.via) methods for cognitive subgroups. Abbreviations: CU, cognitively unimpaired; MCI, mild cognitive impairment; DE, dementia. * denotes *p* < 0.05, ** denotes *p* < 0.01, *** denotes *p* < 0.001 and **** denotes *p* < 0.0001.

**Figure 3 ijms-25-07649-f003:**
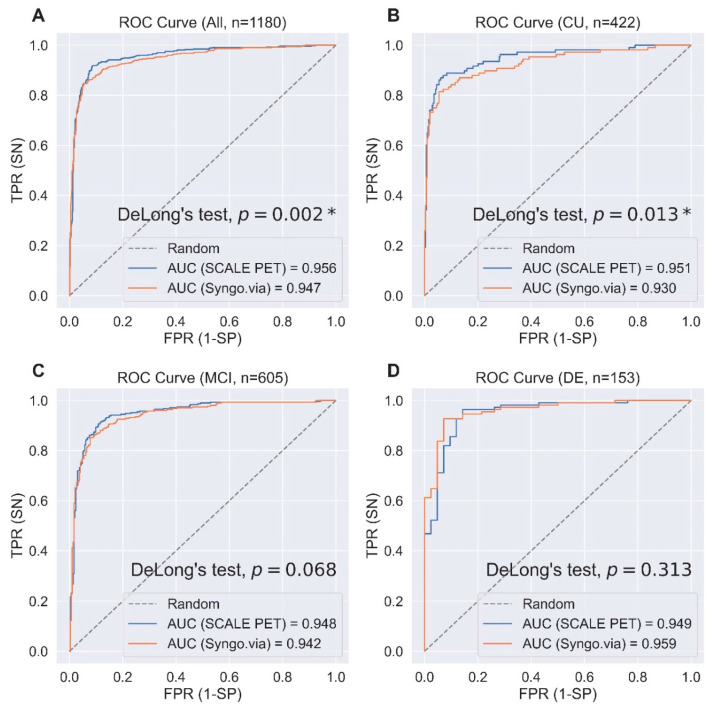
Comparing receiver operating characteristics curves for predicting amyloid positivity with logistic regression models and regional SUVRs from the deep learning-based PET-MRI (SCALE PET) and the PET-only (Syngo.via) methods in all participants group (**A**) and cognitively unimpaired (**B**), mild cognitive impairment (**C**), and dementia (**D**) subgroups. Abbreviations: Receiver operating characteristics, ROC; AUC, Area under the curve; TPR, true positive rate; SN, sensitivity; FPR, false positive rate; SP, specificity; CU, cognitively unimpaired; MCI, mild cognitive impairment; DE, dementia. * denotes *p* < 0.05.

**Figure 4 ijms-25-07649-f004:**
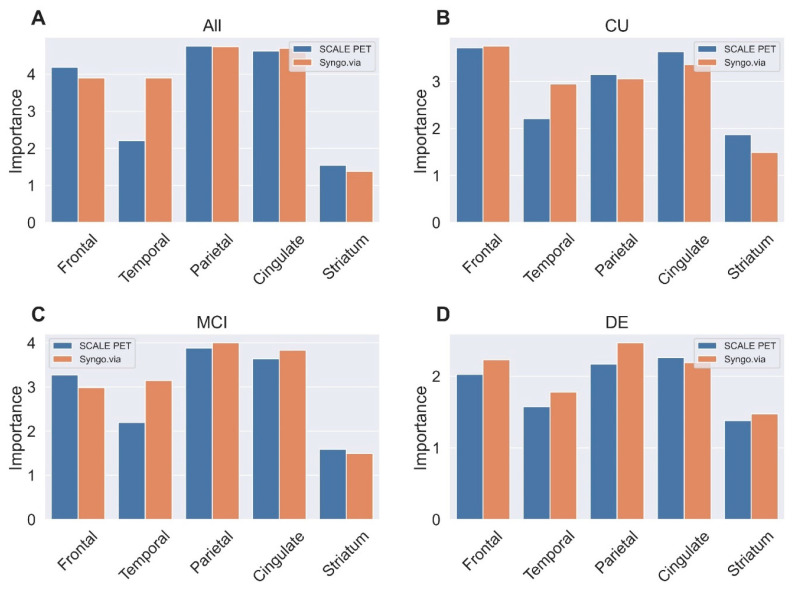
Comparing feature importance of the logistic regression model with regional SUVRs from the deep learning-based PET-MRI (SCALE PET) and the PET-only (Syngo.via) methods for predicting amyloid positivity in all participants group (**A**) and cognitively unimpaired (**B**), mild cognitive impairment (**C**), and dementia (**D**) subgroups. Abbreviations: CU, cognitively unimpaired; MCI, mild cognitive impairment; DE, dementia.

**Figure 5 ijms-25-07649-f005:**
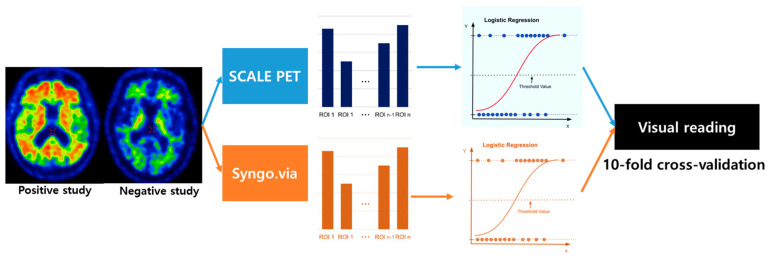
Flow diagram of the PET analysis process.

**Table 1 ijms-25-07649-t001:** Demographic and clinical characteristics of the study participants.

	Amyloid Negative (*n* = 651)	Amyloid Positive (*n* = 539)	Statistics (*p*-Value)
Age, years	72.84 ± 9.40	76.18 ± 7.99	*t* = −6.49 (*p* < 0.001 *)
Education, years	10.71 ± 5.19	10.21 ± 5.40	*t* = 1.61 (*p* = 0.109)
Sex, *n* (%)			χ^2^ = 0.57 (*p* = 0.452)
Female	461 (70.8%)	363 (68.6%)	
Male	190 (29.2%)	166 (31.4%)	
Diagnosis, *n* (%)			χ^2^ = 120.73 (*p* < 0.001 *)
CU	314 (48.2%)	108 (20.4%)	
MCI	295 (45.3%)	310 (58.6%)	
DE	42 (6.5%)	111 (21.0%)	
*APOE*			χ^2^ = 134.18 (*p* < 0.001 *)
ε4 carrier, *n* (%)	127 (19.5%)	274 (51.8%)	
ε4 non-carrier, *n* (%)	524 (80.5%)	255 (48.2%)	
CDR	0.31 ± 0.32	0.58 ± 0.42	*t* = −12.44 (*p* < 0.001 *)
CDR-SB	1.28 ± 1.86	2.83 ± 2.84	*t* = −11.28 (*p* < 0.001 *)

Abbreviations: CU, cognitively unimpaired; MCI, mild cognitive impairment; DE, dementia; CDR, Clinical Dementia Rating; CDR-SB, Clinical Dementia Rating-Sum of Boxes. The data are presented as the “mean ± SD” format for continuous variables and the “counts (proportion in percentage)” format for categorical variables. * denotes *p* < 0.05.

**Table 2 ijms-25-07649-t002:** Predictive performance for deep learning-based PET-MRI (SCALE PET) and PET-only (Syngo.via) methods in predicting amyloid positivity using regional SUVRs.

	Method	Accuracy	Sensitivity	Specificity	F1 Score	AUROC
All	SCALE PET	0.914	0.917	0.912	0.906	0.956
	Syngo.via	0.900	0.845	0.945	0.883	0.947
CU	SCALE PET	0.919	0.861	0.939	0.845	0.951
	Syngo.via	0.910	0.806	0.946	0.821	0.930
MCI	SCALE PET	0.898	0.903	0.892	0.900	0.948
	Syngo.via	0.884	0.848	0.922	0.883	0.942
DE	SCALE PET	0.928	0.955	0.857	0.951	0.949
	Syngo.via	0.922	0.919	0.929	0.944	0.959

Abbreviations: AUROC, area under receiver operating curve; CU, cognitively unimpaired; MCI, mild cognitive impairment; DE, dementia.

## Data Availability

The datasets generated or analyzed during the current study are not publicly available due to the Patient Data Management Protocol of Yeouido St. Mary’s Hospital but are available from the corresponding author upon reasonable request.

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
