# Peer review of "A Comparative Analysis of Two Automated Quantification Methods for Regional Cerebral Amyloid Retention: PET-Only and PET-and-MRI-Based Methods"

_ijms, 2024, doi:10.3390/ijms25147649_

Round 1

Reviewer 1 Report

Comments and Suggestions for Authors

Dear Authors,

I have carefully read your manuscript, which aimed to evaluate the discriminative ability to identify Aβ-positivity using two established quantification software—conventional PET-only method and deep learning-based method using PET and MRI—in a large population including cognitively normal or impaired samples.

The topic is of clinical significance and interest to the readers.

The aim is too long. I suggest to revise it and make it more concise. Also, include it in the Abstract after revision.

Introduction is well written with sufficient references.

In the methods, there is no study type. Please use STROBE criteria and add it. Also add answers to the following questions. When was informed consent signed? What are exclusion criteria? What is the study outcome? Which variables are investigated? How did you find 569 ab positive patients in a single center?

Results are well presented. Figures on imaging details are lacking and should be included.

Discussion should include more references and your personal opinion on study findings.

Comments on the Quality of English Language

Moderate revision is needed

Author Response

Reviewer #1

I have carefully read your manuscript, which aimed to evaluate the discriminative ability to identify Aβ-positivity using two established quantification software—conventional PET-only method and deep learning-based method using PET and MRI—in a large population including cognitively normal or impaired samples. The topic is of clinical significance and interest to the readers.

Response: Thank you so much for taking time to review our manuscript and give detailed and considerate comments, which has greatly helped to improve this paper. The responses to your comments are appended by point-to-point and all the changes are indicated in the revised version with yellow highlights.

Comment 1: The aim is too long. I suggest to revise it and make it more concise. Also, include it in the Abstract after revision. Introduction is well written with sufficient references.

Response 1: Thank you for your comment. We revised the abstract after modifying manuscript as follows:

(The original abstract) “Accurate quantification of amyloid deposition from positron emission tomography (PET) is essential for early detection and intervention in Alzheimer's disease (AD), but there is still a lack of studies comparing the performance among various kinds of automated methods which have their pros and cons. This study compared 1) the PET-only method which registers PET image to standard space for cortical parcellation and segmentation with 2) a more recent method that uses pre-trained deep-learning segmentation model for PET and magnetic resonance image (MRI) in individual space (PET-MRI method). A large sample of 1,180 participants in the Catholic Aging Brain Imaging (CABI) database was analyzed to calculate the regional standardized uptake value ratio (SUVR) using both methods. The logistic regression models were employed to assess the discriminability of amyloid-positive and negative groups through 10-fold cross-validation and area under the receiver operating characteristics metrics. The two methods showed high correlation in calculating SUVRs. However, the PET-MRI method, incorporating MRI data for anatomical accuracy, demonstrated superior performance in predicting amyloid-positivity. The parietal, frontal, and cingulate importantly contributed to the prediction. The PET-MRI method with a pre-trained deep-learning model approach provides an efficient and precise method for earlier diagnosis and intervention in the AD continuum.”

(The revised abstract) “Accurate quantification of amyloid positron emission tomography (PET) is essential for early detection and intervention in Alzheimer's disease (AD), but there is still a lack of studies comparing the performance of various automated methods. This study compared the PET-only method and PET-and-MRI-based method with pre-trained deep-learning segmentation model. A large sample of 1,180 participants in the Catholic Aging Brain Imaging (CABI) database was analyzed to calculate the regional standardized uptake value ratio (SUVR) using both methods. The logistic regression models were employed to assess the discriminability of amyloid-positive and negative groups through 10-fold cross-validation and area under the receiver operating characteristics (AUROC) metrics. The two methods showed high correlation in calculating SUVRs, but the PET-MRI method, incorporating MRI data for anatomical accuracy, demonstrated superior performance in predicting amyloid-positivity. The parietal, frontal, and cingulate importantly contributed to the prediction. The PET-MRI method with a pre-trained deep-learning model approach provides an efficient and precise method for earlier diagnosis and intervention in the AD continuum.”

Comment 2: In the methods, there is no study type. Please use STROBE criteria and add it. Also add answers to the following questions. When was informed consent signed? What are exclusion criteria? What is the study outcome? Which variables are investigated? How did you find 569 ab positive patients in a single center?

Response 2: Thank you for your professional recommendation regarding the use of STROBE criteria. Our study is a cross-sectional study based on the data extracted through retrospective chart review, and the IRB granted waivers of consent for our study. The investigated variables in our study as included demographic variables¾age, sex, educational years, APOE e4 carrier status, and cognitive test results from the CERAD battery¾, imaging variables such as regional and global Standardized Uptake Value Ratios (SUVRs) calculated using Syngo.via and SCALE PET, and an amyloid positivity assessed by nuclear medicine experts as an outcome variable.

Annually, about 17,000 patients visit our clinic with complaints of memory impairment. Of these, around 400 individuals underwent amyloid PET scans. We enrolled patients who had PET scan images and accessible records of the required demographic variables and CERAD-K scores according to the inclusion criteria. This process enabled us to include 569 amyloid-positive patients during the study period from our clinic alone.

We clarified the information about the study design, investigated variables, and inclusion and exclusion criteria more explicitly in the method section according to STROBE criteria as follows:

(L284 at P9 in the revised manuscript) “The inclusion criteria were subjects older than 36 years with an amyloid PET scan and the Korean Version of the Consortium to Establish a Registry for Alzheimer's Disease Assessment Packet (CERAD-K) result rated by trained psychologists. The CERAD-K battery assesses several cognitive domains including memory (word list memory/recall/recognition and constructional praxis recall), visuospatial construction (constructional praxis), language (15-item Boston Naming test), attention, and executive functions (verbal fluency) (Lee et al., 2002). The exclusion criteria were individuals (i) having any psychiatric or neurological disorder except for dementia, or (ii) unstable medical conditions.”

(L306 at P9 in the revised manuscript) “The design of this study is a cross-sectional study based in the data extracted by retrospective chart review, and the IRB granted waivers of consent for the participants. The demographic variables including age, sex, educational years, APOE e4 carrier status, and cognitive test results from the CERAD battery was obtained from the CABI dataset and analyzed in our study.”

Comment 3: Results are well presented. Figures on imaging details are lacking and should be included.

Response 3: Thank you for your positive feedback. We described information about imaging details at subsection “4.2. Amyloid PET and MRI image acquisition”. We added a figure to explain the image acquisition process in the supplementary material (Figure S3).

(L304 at P9 in the original abstract) “Static PET scans were acquired 90 minutes after intravenous injection of 185MBq of 18F-flutemetamol (FMM) for 20 minutes with Biograph 40 TruePoint (Siemens Medical Solutions, Erlangen, Germany). The matrix size was 256 ´ 256 ´ 175 and the voxel size was 1.3364 ´ 1.3364 ´ 3 mm3. Reconstruction of a static image was performed with the 2D-ordered subsets expectation-maximization algorithm with two iterations for 21 subsets. The CT scans were acquired for attenuation correction before the PET scans.

Structural T1-weighted images were scanned through Siemens Skyra 3T scanner (Siemens Healthcare, Erlangen, Germany) and a 20-channel head & neck coil. The magnetization-prepared rapid gradient echo scan (MPRAGE) sequence was utilized with scanning parameters as follows: repetition time (TR) = 1860 ms, echo time (TE) = 25.3 ms, flip angle = 9°, field-of-view (FOV) = 224 x 224 mm, matrix size of 256 × 256, 208 axial slices with a slice thickness of 1.0 mm. All acquired Digital Imaging and Communications in Medicine (DICOM) image files were anonymized and converted to NIfTI format with “dcm2niix” software.”

Figure S3. Detailed imaging acquisition process.

Comment 4: Discussion should include more references and your personal opinion on study findings.

Response 4: Thank you for your advice. We added opinions and references to the discussion as follows.

(L255 at P8 in the revised manuscript) “The superior performance of the PET-MRI method in cognitively normal subjects has significant implications. This is because the importance of early diagnosis is increasing in the AD field, where diagnostic framework based on biomarkers is evolving and DMTs are being introduced. DMTs have been approved for use not at the dementia stage, where neuronal damage has already advanced, but at the MCI or mild dementia stage, and only in patients who have been confirmed to be Aβ-positive (U.S. Food and Drug Administration, 2023; Van Dyck et al., 2023). Recently, researchers have argued that the indications for DMTs should be extended to preclinical AD (Rafii et al., 2023), based on the A-T-N (Amyloid-tau-neurodegeneration) hypothesis (Jack et al., 2010). Considering the pathological characteristics where accumulation of tau protein and neurodegeneration progress after Aβ accumulation in the brain, it is believed that more aggressive treatment at the stage where Aβ begins to accumulate can prevent cognitive decline by halting the progression to the next pathological stage (He et al., 2018; Sperling, Mormino, & Johnson, 2014). To administer treatment from the early stages of AD, even before symptoms appear, it is essential to identify amyloid-beta positivity in the normal population to select the appropriate subjects for treatment (Verger et al., 2023). Therefore, the PET-MRI method, which can better diagnose AD in cognitively unimpaired subjects, could play a crucial role in such proactive early treatment and might potentially replace the PET-only method in the future.”

Reviewer 2 Report

Comments and Suggestions for Authors

Minor english correction required.

Comments on the Quality of English Language

Sunghwan Kim et al presents an important study comparing two methods for quantifying amyloid deposition using PET imaging for Alzheimer’s disease detection. The authors needs to work on these comments below:

1)     The author needs to generalize the findings to different demographic and ethnic groups as this study is now based on CABI database which make its potential bias

Author Response

Point-by-point Response to Reviewer’s Comments

Reviewer #2

Sunghwan Kim et al presents an important study comparing two methods for quantifying amyloid deposition using PET imaging for Alzheimer’s disease detection. The authors needs to work on these comments below:

Response: Thank you very much for taking time to review our manuscript and give comments. The responses to your comments are appended by point-to-point and all the changes are indicated in the revised version with yellow highlights.

Comment 1The author needs to generalize the findings to different demographic and ethnic groups as this study is now based on CABI database which make its potential bias.

Response 1: We appreciate your comment and agree that there would be concern about limited generalizability because our study population is recruited from a single memory clinic, resulting in potentially very homogeneous demographic characteristics, including ethnicity. We clarified this limitation in our manuscript as below:

(L287 at P9 in the original manuscript) “The limitations of our study include the following: firstly, it is a single-center study, using only one reference ROI and tracer. Secondly, the ROIs between the two methods did not precisely match, leading to potential bias from ROI selection. Sensitivity analysis with multiple tracers, multiple centers, and multiple reference and target ROIs should be followed in the future to address these issues.”

 (L270 at P8 in the revised manuscript) “The limitations of our study include the following: Firstly, it is a single-center study, using only one reference ROI and tracer. Therefore, the generalizability of our results remains questionable. Multi-center studies using multiple tracers with different reference ROIs should be followed in the future. Secondly, the ROIs between the two methods did not precisely match, leading to potential bias from ROI selection. Sensitivity analysis with varying target ROIs should be conducted to address these issues.”

Once again, thank you so much for your valuable comments and advice which contributed significantly to the revision of our manuscript. I hope the revised manuscript can meet the requirements of International Journal of Molecular Sciences for publication. If there are any further changes required, we would be more than happy to make correction to meet your standard.

Reviewer 3 Report

Comments and Suggestions for Authors

A comparative analysis of two automated quantification methods for regional cerebral amyloid retention: PET-only and PET- and-MRI-based methods

I read the manuscript with interest. The authors can find my suggestions, recommendations, and appraisal section by section as follows:

Introduction: Please, remove the first sentence from the main text.

The first paragraph of the introduction is well written and contains all the info relevant to introducing β-amyloid and AD.  Moreover, the section explains and introduces fluidly the topic and hypotheses, which are very easy to follow. Despite, this I suggest adding a brief description of the main symptoms of AD and premorbid conditions in the first or second paragraph. Moreover, I suggest to add also a brief explanation or more info about amyloid PET. The present is a journal of molecular sciences, and I mean that the readers are interested about the technical/ molecular bases of amyloid PET. Despite this, the section is written in an extremely clear and rigorous way.

4. Materials and Methods: More info about CERAD-K is needed. Please, add. Maybe, 18F, but I think that is the same.

4.3. Visual analysis of Amyloid PET: Please, specify if the analysis was or not independently performed.

Statistical analyses: Line 351- “Demographic and clinical characteristics were compared employing independent t- 351 tests and chi-square tests.” I do not agree with the use of t-tests, a MANOVA, with follow-up ANOVAs is better. However, Did you apply any correction to Chi Sqr? Similarly, it is not clear RMSE. Please clarify this sentence. Similarly, more info about SVM is needed (you can add to supplementary materials).

The Results are interesting, and the accuracy showed interestingly lower values for MCI (confirmed in the supplementary materials).

The Discussion is well-written and integrates the results with the existing literature. 

Author Response

Point-by-point Response to Reviewer’s Comments

Reviewer #3

I read the manuscript with interest. The authors can find my suggestions, recommendations, and appraisal section by section as follows:

Response: Thank you so much for taking time to review our manuscript and give detailed and considerate comments, which has greatly helped to improve this paper. The responses to your comments are appended by point-to-point and all the changes are indicated in the revised version with yellow highlights.

Comment 1: Introduction: Please, remove the first sentence from the main text.

Response 1: We apologize for our mistake we retained the first sentence originally written in the journal’s template for the manuscript. We removed that sentence which is not relevant to our study.

Comment 2: The first paragraph of the introduction is well written and contains all the info relevant to introducing β-amyloid and AD. Moreover, the section explains and introduces fluidly the topic and hypotheses, which are very easy to follow. Despite, this I suggest adding a brief description of the main symptoms of AD and premorbid conditions in the first or second paragraph. Moreover, I suggest to add also a brief explanation or more info about amyloid PET. The present is a journal of molecular sciences, and I mean that the readers are interested about the technical/ molecular bases of amyloid PET. Despite this, the section is written in an extremely clear and rigorous way.

Response 2: Thank you for your insightful comments. We included brief explanation about clinical symptoms of AD and amyloid PET as below:

 (L42 at P1 in the revised manuscript) “The Alzheimer’s disease (AD) is the most common cause of dementia thus one of most burdensome disease globally (Collaborators, 2021). Individuals affected by the disease exhibit a wide range of symptoms, such as memory impairment, behavioral disturbance and psychological symptoms (Joe & Ringman, 2019; Lyketsos et al., 2011), or may even lack any clinical symptoms, which is called as preclinical AD (Dubois et al., 2016). Recently, the importance of biomarkers measuring the deposition of β-amyloid (Aβ) plaques in the brain—a central pathological hallmark of AD—has increasingly been recognized (Dubois, von Arnim, Burnie, Bozeat, & Cummings, 2023; Zetterberg & Bendlin, 2021).”

(L61 at P2 in the revised manuscript) “Amyloid PET scan using radioactive tracers (details are described in the supplementary materials) which bind to fibrillar amyloid aggregates in the brain allows physicians to enable in vivo visualization of spatial distribution and regional quantification of Aβ protein deposits in the brain (Lowe et al., 2019; Thal et al., 2015). It reflects a more cumulative effect of Aβ and the static status of AD compared to fluid biomarkers (Ben Bouallègue, Mariano-Goulart, Payoux, & the Alzheimer’s Disease Neuroimaging, 2017; Olsson et al., 2016).”

(P1 at the “Amyloid positron emission topography” subsection of the supplementary methods in the revised supplementary material) “Amyloid positron emission topography (PET) is an imaging technique used to assess the abnormal deposition of β-amyloid (Aβ) in the brain. The PET scanner detects the level of absorbed radioactive ligand in living cells or tissues, which is intravenously administered to the subject before scanning. This imaging technique has a beneficial characteristic that it enables molecule-specific in vivo visualization in humans or animals. To identify the core pathology of Alzheimer’s disease (AD), Aβ-specific ligand should be used. The first radioactive ligand was the 11C-labeled Pittsburgh compound B (11C-PiB). Its performance of predicting amyloid-positivity was confirmed by autopsy study, but its clinical use was limited due to the short half-life of 11C radioisotope, which is only 20 minutes. To overcome this drawback, several fluorime-18(18F)-derived tracers with a longer half-life (110 minutes) have been developed.

Currently, three 18F amyloid PET tracers are currently available for clinical application and have been validated as the gold standard; 18F-florbetapir (Amyvid™; Avid Radiopharmaceuticals), 18F-florbetapir (Amyvid™; Avid Radiopharmaceuticals; approved in 2012), 18F-flutemetamol (Vizamyl™; GE Healthcare; approved in 2013), and 18F-florbetaben (Neuraceq™; Life Molecular Imaging; approved in 2014). Each molecule has a distinct chemical structure, resulting in different pharmacokinetic and binding properties. All three tracers have been granted by the Food and Drug Administration (FDA) and European Medicines Authority (EMA) for clinical use, and they also received local regulatory approval in South Korea. In this study, we used 18F-flutemetamol as routine clinical practice.”

Comment 3: 4. Materials and Methods: More info about CERAD-K is needed. Please, add. Maybe, 18F, but I think that is the same.

Response 3: Thank you very much for your detailed guidance. We added detailed information about CERAD-K and modified the description for the tracer.

(L301 at P9 in the revised manuscript) “The inclusion criteria were subjects older than 36 years with an amyloid PET scan and the Korean Version of the Consortium to Establish a Registry for Alzheimer's Disease Assessment Packet (CERAD-K) result rated by trained psychologists. The CERAD-K battery assesses several cognitive domains including memory (word list memory/recall/recognition and constructional praxis recall), visuospatial construction (constructional praxis), language (15-item Boston Naming test), attention, and executive functions (verbal fluency) (Lee et al., 2002). The exclusion criteria were individuals (i) having any psychiatric or neurological disorder except for dementia, or (ii) unstable medical conditions.”

(L331 at P9 in the revised manuscript) “Static PET scans were acquired 90 minutes after intravenous injection of 185MBq of 18F-flutemetamol (FMM) for 20 minutes with Biograph 40 TruePoint (Siemens Medical Solutions, Erlangen, Germany).”

Comment 4: 4.3. Visual analysis of Amyloid PET: Please, specify if the analysis was or not independently performed.

Response 4: Thank you for your advice. We revised the sentence specifying the independence of visual results as below:
(P9 in the original manuscript) “The visual reading results were used as a gold standard for Aβ-positivity.”

 (L352 at P10 in the revised manuscript) “The visual reading results were obtained independently of the quantification results and were used as a gold standard for Aβ-positivity.”

Comment 5: Statistical analyses: Line 351- “Demographic and clinical characteristics were compared employing independent t-tests and chi-square tests.” I do not agree with the use of t-tests, a MANOVA, with follow-up ANOVAs is better. However, Did you apply any correction to Chi Sqr? Similarly, it is not clear RMSE. Please clarify this sentence. Similarly, more info about SVM is needed (you can add to supplementary materials).

Response 5: Thank you for your guidance. In terms of statistical tests for demographical variables, MANOVA or ANOVA might be worth considering. However, we thought that the independent t-test is appropriate for comparing the means of continuous univariate data, and the chi-square test is proper to test the difference in frequency of categorical univariate data between two distinct groups, each consisting of more than 500 participants. To our knowledge, there are no applicable corrections for the chi-square test in this context, thus we would appreciate further comments on any additional aspects that should be analyzed in more detail.

We clarified the sentence explaining RMSE in revised version of manuscript and the description for SVM in the supplementary materials as follows:

(L381 at P10 in revised version of manuscript) “To be specific, the sum of the squared differences between the SUVRs obtained from the two methods was divided by the number of samples in each group, and the square root of the result was taken to calculate the RMSE.”

(P1 at the “Amyloid-positivity prediction models” subsection of the supplementary methods in the revised supplementary material) “The logistic regression (LR) and support vector machine (SVM) models were implemented to predict visual readings of amyloid positron emission tomography (PET) assessed by nuclear medicine experts. The LR model is a prediction model that uses a linear combination of the Standardized Uptake Value Ratios (SUVRs) and applies logit function to calculate the probability of belonging to amyloid positive or negative group. During fitting process, the model determines the coefficients of linear combination to minimize prediction errors. The SVM model, on the other hand, finds a separating hyper-plane with maximum margin between groups to classify the subjects during the fitting process. Because of their transparency and simplicity, these classic supervised machine learning algorithms have widely used for a variety of classification task and showed reliable and robust discriminability. To test reproducibility and model-independence of the LR models’s ability to discriminate between amyloid-positive and negative subjects, we additionally implemented the linear SVM model (with default model parameters specified in the scikit-learn python library) for the same task described in the method section of the main manuscript.”

Comment 6: The Results are interesting, and the accuracy showed interestingly lower values for MCI (confirmed in the supplementary materials). The Discussion is well-written and integrates the results with the existing literature. 

Response 6: Thank you for your insights regarding the results. As you noted, there are some differences in performance metrics among subgroups (CU, MCI, and DE) in the Table 2. In the aspect of accuracy, the two quantification methods (SCALE PET and Syngo.via) seem to be less accurate in the MCI subgroup (0.898 for SCALE PET and 0.884 for Syngo.via) compared to the CU or DE subgroups (0.919 for SCALE PET and 0.910 for for Syngo.via in CU; 0.928 for SCALE PET and 0.922 for for Syngo.via in DE). However, the AUROC showed relatively consistent results across subgroups. There has been debate about which metric is better for assessing performance of classification models, and several studies have empirically demonstrated that the accuracy results can sometimes be biased, whereas AUROC is considered as a robust measure for various distributions and populations (Jin & Ling, 2005). Therefore, many machine-learning experts recommend that the accuracy of classifiers should be accompanied by other metrics, such as AUROC. Consequently, we displayed various performance metrics and based our main results on the AUROC for each classification model.

Once again, thank you so much for your valuable comments and advice which contributed significantly to the revision of our manuscript. I hope the revised manuscript can meet the requirements of International Journal of Molecular Sciences for publication. If there are any further changes required, we would be more than happy to make correction to meet your standard.

References

Ben Bouallègue, F., Mariano-Goulart, D., Payoux, P., & the Alzheimer’s Disease Neuroimaging, I. (2017). Comparison of CSF markers and semi-quantitative amyloid PET in Alzheimer’s disease diagnosis and in cognitive impairment prognosis using the ADNI-2 database. Alzheimer's Research & Therapy, 9(1), 32. doi:10.1186/s13195-017-0260-z

Collaborators, G. B. D. (2021). Global mortality from dementia: Application of a new method and results from the Global Burden of Disease Study 2019. Alzheimers Dement (N Y), 7(1), e12200. doi:10.1002/trc2.12200

Dubois, B., Hampel, H., Feldman, H. H., Scheltens, P., Aisen, P., Andrieu, S., . . . Blennow, K. (2016). Preclinical Alzheimer's disease: definition, natural history, and diagnostic criteria. Alzheimer's & dementia, 12(3), 292-323.

Dubois, B., von Arnim, C. A. F., Burnie, N., Bozeat, S., & Cummings, J. (2023). Biomarkers in Alzheimer’s disease: role in early and differential diagnosis and recognition of atypical variants. Alzheimer's Research & Therapy, 15(1), 175. doi:10.1186/s13195-023-01314-6

Jin, H., & Ling, C. X. (2005). Using AUC and accuracy in evaluating learning algorithms. IEEE Transactions on Knowledge and Data Engineering, 17(3), 299-310. doi:10.1109/TKDE.2005.50

Joe, E., & Ringman, J. M. (2019). Cognitive symptoms of Alzheimer’s disease: clinical management and prevention. bmj, 367.

Lee, J. H., Lee, K. U., Lee, D. Y., Kim, K. W., Jhoo, J. H., Kim, J. H., . . . Woo, J. I. (2002). Development of the Korean version of the Consortium to Establish a Registry for Alzheimer's Disease Assessment Packet (CERAD-K): clinical and neuropsychological assessment batteries. J Gerontol B Psychol Sci Soc Sci, 57(1), P47-53. doi:10.1093/geronb/57.1.p47

Lowe, V. J., Lundt, E. S., Albertson, S. M., Przybelski, S. A., Senjem, M. L., Parisi, J. E., . . . Murray, M. E. (2019). Neuroimaging correlates with neuropathologic schemes in neurodegenerative disease. Alzheimers Dement, 15(7), 927-939. doi:10.1016/j.jalz.2019.03.016

Lyketsos, C. G., Carrillo, M. C., Ryan, J. M., Khachaturian, A. S., Trzepacz, P., Amatniek, J., . . . Miller, D. S. (2011). Neuropsychiatric symptoms in Alzheimer’s disease. In (Vol. 7, pp. 532-539): Elsevier.

Olsson, B., Lautner, R., Andreasson, U., Öhrfelt, A., Portelius, E., Bjerke, M., . . . Strobel, G. (2016). CSF and blood biomarkers for the diagnosis of Alzheimer's disease: a systematic review and meta-analysis. The lancet neurology, 15(7), 673-684. Retrieved from https://www.thelancet.com/journals/laneur/article/PIIS1474-4422(16)00070-3/abstract

https://www.sciencedirect.com/science/article/pii/S1474442216000703?via%3Dihub

Thal, D. R., Beach, T. G., Zanette, M., Heurling, K., Chakrabarty, A., Ismail, A., . . . Buckley, C. (2015). [(18)F]flutemetamol amyloid positron emission tomography in preclinical and symptomatic Alzheimer's disease: specific detection of advanced phases of amyloid-β pathology. Alzheimers Dement, 11(8), 975-985. doi:10.1016/j.jalz.2015.05.018

Zetterberg, H., & Bendlin, B. B. (2021). Biomarkers for Alzheimer’s disease—preparing for a new era of disease-modifying therapies. Molecular Psychiatry, 26(1), 296-308. doi:10.1038/s41380-020-0721-9

Round 2

Reviewer 1 Report

Comments and Suggestions for Authors

Thank you for revising the paper according to the suggestions. The paper's quality has improved.

Reviewer 2 Report

Comments and Suggestions for Authors

Thanks

Reviewer 3 Report

Comments and Suggestions for Authors

The manuscript has been improved as l requested.